# Mesothelin Expression in Human Tumors: A Tissue Microarray Study on 12,679 Tumors

**DOI:** 10.3390/biomedicines9040397

**Published:** 2021-04-07

**Authors:** Sören Weidemann, Pauline Gagelmann, Natalia Gorbokon, Maximilian Lennartz, Anne Menz, Andreas M. Luebke, Martina Kluth, Claudia Hube-Magg, Niclas C. Blessin, Christoph Fraune, Katharina Möller, Christian Bernreuther, Patrick Lebok, Till S. Clauditz, Frank Jacobsen, Jakob R. Izbicki, Kristina Jansen, Guido Sauter, Ria Uhlig, Waldemar Wilczak, Stefan Steurer, Sarah Minner, Eike Burandt, Rainer H. Krech, David Dum, Till Krech, Andreas H. Marx, Ronald Simon

**Affiliations:** 1Institute of Pathology, University Medical Center Hamburg-Eppendorf, 20357 Hamburg, Germany; s.weidemann@uke.de (S.W.); pauline.gagelmann@gmail.com (P.G.); n.gorbokon@uke.de (N.G.); m.lennartz@uke.de (M.L.); a.menz@uke.de (A.M.); luebke@uke.de (A.M.L.); m.kluth@uke.de (M.K.); c.hube@uke.de (C.H.-M.); n.blessin@luke.de (N.C.B.); c.fraune@uke.de (C.F.); ka.moeller@uke.de (K.M.); c.bernreuther@uke.de (C.B.); p.lebok@uke.de (P.L.); t.clauditz@uke.de (T.S.C.); f.jacobsen@uke.de (F.J.); g.sauter@uke.de (G.S.); r.uhlig@uke.de (R.U.); w.wilczak@uke.de (W.W.); s.steurer@uke.de (S.S.); s.minner@uke.de (S.M.); e.burandt@uke.de (E.B.); d.dum@uke.de (D.D.); t.krech@uke.de (T.K.); Andreas.Marx@klinikum-fuerth.de (A.H.M.); 2General, Visceral and Thoracic Surgery Department and Clinic, University Medical Center Hamburg-Eppendorf, 20357 Hamburg, Germany; izbicki@uke.de (J.R.I.); k.jansen@uke.de (K.J.); 3Institute of Pathology, Clinical Center Osnabrueck, 49074 Osnabrueck, Germany; pathologie@ncid.net; 4Department of Pathology, Academic Hospital Fuerth, 90766 Fuerth, Germany

**Keywords:** mesothelin, multi-tumor tissue micro array, immunohistochemistry

## Abstract

Mesothelin (MSLN) represents an attractive molecule for targeted cancer therapies. To identify tumors that might benefit from such therapies, tissue microarrays including 15,050 tumors from 122 different tumor types and 76 healthy organs were analyzed for MSLN expression by immunohistochemistry. Sixty-six (54%) tumor types showed at least occasional weak staining, including 50 (41%) tumor types with at least one strongly positive sample. Highest prevalence of MSLN positivity had ovarian carcinomas (serous 97%, clear cell 83%, endometrioid 77%, mucinous 71%, carcinosarcoma 65%), pancreatic adenocarcinoma (ductal 75%, ampullary 81%), endometrial carcinomas (clear cell 71%, serous 57%, carcinosarcoma 50%, endometrioid 45%), malignant mesothelioma (69%), and adenocarcinoma of the lung (55%). MSLN was rare in cancers of the breast (7% of 1138), kidney (7% of 807), thyroid gland (1% of 638), soft tissues (0.3% of 931), and prostate (0 of 481). High expression was linked to advanced pathological tumor (pT) stage (*p* < 0.0001) and metastasis (*p* < 0.0001) in 1619 colorectal adenocarcinomas, but unrelated to parameters of malignancy in 1072 breast-, 386 ovarian-, 174 lung-, 757 kidney-, 171 endometrial-, 373 gastric-, and 925 bladder carcinomas. In summary, numerous important cancer types with high-level MSLN expression might benefit from future anti-MSLN therapies, but MSLN’s prognostic relevance appears to be limited.

## 1. Introduction

The mesothelin (MSLN) gene, located at chromosome 16p13.3, encodes for a membranous precursor glycoprotein that is subsequently cleaved into the soluble 31kD protein megakaryocyte potentiating factor (MPF) and the 40kD membrane-bound protein MSLN [1,2,3]. MSLN was first described as a membrane protein expressed in normal and neoplastic mesothelial cells, but subsequent studies demonstrated a broader expression pattern [1,4,5,6,7,8,9]. The function of MSLN is not fully understood. In normal cells, MSLN does not seem to be essential as a homozygous MSLN mutant mouse lacking MSLN protein developed and reproduced normally [10]. However, MSLN has been identified as a specific binding protein of cancer antigen (CA125) that mediates cell adhesion [11,12]. This interaction was suggested to play a role in the development of peritoneal metastasis [11,13]. In cell line and animal experiments, MSLN overexpression was shown to activate the PI3K/AkT, NFκB, and MAPK/ERK pathways, to hinder apoptosis and to promote cell proliferation, migration, and metastasis [14,15,16,17,18,19,20,21].

MSLN is expressed in only few normal tissues but has been found to be overexpressed in various tumor types at a relevant frequency [4,5,6,7,8,9,22]. Therefore, and due to its membranous location, MSLN represents an attractive molecule for target-specific cancer therapies. Targeted therapies use drugs to inhibit specific genes and proteins that are involved in tumor cell growth. Several therapeutic approaches, including adaptive immunotherapy (CAR-T cells, TC-210 T cells), monoclonal antibodies (Amatuximab/MORAb-009), recombinant immunotoxins (SS1P, LMB-100/RG7787), antibody-drug conjugates (Anetumab Ravtansine/BAY94-9343, DMOT4039A, BAY2287411, BMS-986148, h7D9.v3), listeria monocytogene induced anti-tumor immune response (CRS-207, JNJ-64041757), and immunocytokines (IL12-SS1) have provided encouraging data in animal models and/or clinical phase I and II trials [23,24,25,26,27,28,29,30,31,32,33,34,35,36]. Which tumor entities might benefit most from anti-MSLN therapies is difficult to predict since the literature on MSLN expression is controversial for many tumor entities. For example, MSLN positivity has been described in 38% to 69% of lung adenocarcinomas [37,38,39], 17% to 100% of pancreatic adenocarcinomas [40,41], 2% to 68% of colorectal carcinomas [5,9,42], 55% to 100% of serous carcinomas of the ovary [43,44], 21% to 78% of gastric adenocarcinomas [5,9,45], and 3% to 36% of breast carcinomas [9,46,47]. These conflicting data are likely to be caused by the use of different antibodies, immunostaining protocols, and criteria to categorize MSLN immunostaining in these studies.

To better understand the prevalence and significance of MSLN expression in cancer, a comprehensive study analyzing a large number of neoplastic and non-neoplastic tissues under highly standardized conditions is needed. Therefore, MSLN expression was analyzed in more than 15,000 tumor tissue samples from 122 different tumor types and subtypes, as well as 76 non-neoplastic tissue categories, by immunohistochemistry (IHC) in a tissue microarray (TMA) format in this study.

## 2. Material and Methods

### 2.1. Tissue Microarrays (TMAs)

The normal tissue TMA was composed of 8 samples from 8 different donors for each of 76 different normal tissue types (608 samples on one slide). The cancer TMAs contained a total of 15,050 primary tumors from 122 tumor types and subtypes. Detailed histopathological data on grade, pathological tumor stage (pT), or pathological lymph node status (pN) were available from 7625 tumors (cancers of the colon, bladder, ovarian, endometrium, lung, stomach, breast, and kidney tumors). Clinical follow-up data were available from 1178 breast cancer, 865 kidney cancer, and 254 bladder cancer patients with a median follow-up time of 49/39/14 months (range 1–88/1–250/1–77). No data on previous therapies were available. The composition of both normal and cancer TMAs is described in detail in the results section. All samples were from the archives of the Institutes of Pathology, University Hospital of Hamburg, Germany, the Institute of Pathology, Clinical Center Osnabrueck, Germany, and Department of Pathology, Academic Hospital Fuerth, Germany. Tissues were fixed in 4% buffered formalin and then embedded in paraffin. The TMA manufacturing process was described earlier in detail [48,49]. In brief, one tissue spot (diameter: 0.6 mm) was transmitted from a cancer containing donor block (≥70% cancer cells) in an empty recipient paraffin block. The use of archived remnants of diagnostic tissues for manufacturing of TMAs and their analysis for research purposes, as well as patient data analysis, has been approved by local laws (HmbKHG, §12) and by the local ethics committee (Ethics commission Hamburg, WF-049/09, 25 January 2010). All work has been carried out in compliance with the Helsinki Declaration.

### 2.2. Immunohistochemistry (IHC)

Freshly prepared TMA sections were immunostained on one day in one experiment. Slides were deparaffinized with xylol, rehydrated through a graded alcohol series, and exposed to heat-induced antigen retrieval for 5 min in an autoclave at 121 °C in pH 9 DakoTarget Retrieval Solution^TM^ (Agilent, Santa Clara, CA, USA; #S2367). Endogenous peroxidase activity was blocked with Dako Peroxidase Blocking Solution^TM^ (Agilent, CA, USA; #52023) for 10 min. Primary antibody specific against MSLN protein (mouse monoclonal, MSVA-235, MS Validated Antibodies, Hamburg, Germany) was applied at 37 °C for 60 min at a dilution of 1:150. Bound antibody was then visualized using the EnVision Kit^TM^ (Agilent, CA, USA; #K5007) according to the manufacturer’s directions. The sections were counterstained with hemalaun. A trained pathologist scored all tissue spots and marked tissue spots with questionable findings for revision by a second pathologist. For normal tissues, the staining intensity of positive cells was semi-quantitively recorded (+, ++, +++). For tumor tissues, the percentage of MSLN positive tumor cells was estimated and the staining intensity was semi-quantitatively recorded (0, 1+, 2+, 3+). For statistical analyses, the staining results were categorized into four groups as follows: Negative: no staining at all, weak staining: staining intensity of 1+ in ≤70% or staining intensity of 2+ in ≤30% of tumor cells, moderate staining: staining intensity of 1+ in >70%, staining intensity of 2+ in > 30% but in ≤70% or staining intensity of 3+ in ≤30% of tumor cells, strong staining: staining intensity of 2+ in >70% or staining intensity of 3+ in >30% of tumor cells.

### 2.3. Statistics

Statistical calculations were performed with JMP 14 software (SAS Institute Inc., Cary, NC, USA). Contingency tables and the chi²-test were performed to search for associations between MSLN and tumor phenotype. Survival curves were calculated according to Kaplan-Meier. The Log-Rank test was applied to detect significant differences between groups. A *p*-value of ≤0.05 was defined as significant.

## 3. Results

### 3.1. Technical Issues

A total of 12,679 (84.2%) of 15,050 tumor samples were interpretable in the TMA analysis. The remaining 2371 (15.8%) samples were not analyzable due to the lack of unequivocal tumor cells or loss of the tissue spot during the technical procedures. On the normal tissue TMA, a sufficient number of samples was always interpretable per tissue to determine MSLN expression.

### 3.2. MSLN Immunostaining in Normal Tissues

In normal tissues, the strongest MSLN expression was observed in the squamous epithelium of tonsil crypts (Figure 1A), where a fraction of cells (intermediate to superficial cell layers) showed strong (+++) MSLN staining. Strong MSLN immunostaining was also seen in some cells and cell groups of the rectal mucosa (+++), the anal transitional epithelium (+++) (Figure 1B) where staining was often particularly prominent in superficial mucinous cells, amnion cells (+++) (Figure 1C) and some chorion cells (++) of the mature placenta, and some elements of corpuscles of Hassall’s of the thymus (++). A somewhat weaker MSLN staining was seen in scattered cells and groups of cells of endocervical mucosa (++) and endometrium (++), epithelial cells of fallopian tube (apical cell border and cilia; ++) (Figure 1D), some intermediate (neck) cells of the stomach antrum (+), some scattered glands in sublingual (Figure 1E) and Brunner glands, few cells of respiratory epithelium (goblet cells;++), some cells or groups of cells in bronchial glands (++), seminal vesicle (+; not in all samples), and in the cytoplasm of few cells of the adenohypophysis (+). MSLN immunostaining was absent in endothelium and media of the aorta, heart muscle, striated muscle, tongue muscle, myometrium of the uterus, muscular wall of the appendix, esophagus, ileum, kidney pelvis, and urinary bladder, corpus spongiosum of the penis, ovarian stroma, fat, skin, hair follicles and sebaceous glands of the skin, non-keratinizing squamous epithelium from the lip, oral cavity, ectocervix, and the esophagus, urothelium of the kidney pelvis and urinary bladder, spleen, antrum and corpus of the stomach, gallbladder epithelium, liver, kidney (Figure 1F), epididymis, testis, lung, decidua cells, cerebellum, cerebrum, salivary glands, prostate, breast, adrenal gland, and lymphatic tissue.

### 3.3. MSLN Immunostaining in Neoplastic Tissues

A significant MSLN immunostaining was observed in 2413 (19.0%) of 12,679 analyzable tumors, including 8.0% with weak, 3.4% with moderate, and 7.6% with strong staining intensity. The staining pattern was variable. Most positive tumors showed a predominantly apical membranous MSLN staining, often accompanied by a less intense cytoplasmic coloration. Other tumors showed a pure membranous staining or a predominantly cytoplasmic positivity. Representative images are shown in Figure 2. At least an occasional weak MSLN positivity was detected in 66 of 122 (54.1%) different tumor types and tumor subtypes and 50 (41.0%) tumor types and subtypes had at least one tumor sample exhibiting strong positivity. The highest frequencies of MSLN positivity were seen in different subtypes of ovarian (65% to 97%) and endometrium (45% to 71%) carcinomas, pancreatic adenocarcinoma (75% and 81%), malignant mesothelioma (69%), and adenocarcinoma of the lung (55%). Rare or absent MSLN positivity was observed in different subtypes of breast tumors (0% in 50 phyllodes, 0.9% in 294 lobular, 5.3% in 27 tubular, 6.6% in 1391 invasive no special type, and 10.9% in 58 mucinous subtypes), renal cell (0% in 177 oncocytomas, 1.6% in 131 chromophobe, 6.6% in 858 clear cell, and 8.7% in 255 papillary subtypes), and thyroid carcinomas (0% to 1.4%), as well as soft tissue tumors (0.3%), and adenocarcinomas of the prostate (0%). A detailed description of the immunostaining results is given in Table 1 and Figure 3.

### 3.4. MSLN Immunostaining, Tumor Phenotype, and Prognosis

A comparison of MSLN immunostaining with pT, pN, and histological grade in 1619 colorectal adenocarcinomas, 1072 invasive breast carcinomas of no special type, 386 serous carcinomas of the ovary, 174 lung adenocarcinomas, 757 clear cell renal cell carcinomas, and 171 endometrioid endometrial, in 373 gastric, and in 925 bladder carcinomas revealed only a statistically significant association between MSLN immunostaining and pT stage, as well as pN status in colorectal cancer (*p* < 0.0001 each, Table 2). MSLN immunostaining was unrelated to overall survival in 227 bladder carcinomas (*p* = 0.3302; only pT ≥ 2), 593 invasive breast carcinomas of no special types (*p* = 0.0976), and 502 clear cell renal cell carcinomas (*p* = 0.3144, Figure 4). A significant association was found between positive MSLN immunostaining and RAS mutations in colorectal carcinomas (*p* = 0.0010), and triple negative invasive breast carcinomas of no special type (*p* < 0.0001, Table 2). There was also a strong tendency towards higher MSLN expression in HPV positive than in HPV negative squamous cell carcinomas (*p* = 0.0098, Table 3).

## 4. Discussion

The result of our normal tissue analysis for MSLN expression is consistent with a high potential of this protein as a therapeutic target. It is obvious that the risk for potential side effects of targeted therapies is connected to the target protein’s site and level of expression in normal tissues. MSLN expression was mostly seen in organs that are not vital for adult or old-aged people, such as tonsil, thymus, gallbladder, seminal vesicle, fallopian tube, uterus, and placenta. In the stomach, duodenum, rectum, and the anal canal, the fraction of MSLN expressing cells was so low that critical side effects might not occur, even if these rare cells were disturbed by drug effects. Our normal tissue results are largely consistent with RNA expression data derived from the FANTOM5 project [50,51] and the Genotype-Tissue Expression (GTEx) project [52] which are all summarized in the protein atlas (https://www.proteinatlas.org/ENSG00000102854-MSLN/tissue, accessed on 6 January 2021). These RNA data would suggest the lung as the organ, which might be most endangered by side effects of anti-MSLN therapies. A high MSLN RNA expression in the lung could be explained by strong MSLN immunostaining in goblet cells of respiratory epithelium and occasional positivity of bronchial glands. As the alveolar system did not show any MSLN expression, we would expect that possible lung side effects derived from therapeutic anti-MSLN antibodies would rather affect the bronchial system than the alveolar space.

The successful analysis of more than 12,000 tumors identified various cancer types that might be particularly well suited for anti-MSLN drugs. Although the usage of tissue microarrays has disadvantages connected to the small size of the analyzed tissue spots (0.6 mm in diameter), including the risk of missing “relevant” tumor components in heterogenous tumors, or the impossibility to capture different tumor compartments (such as invasion front and tumor center) within one tissue spot, it allows for an unprecedented degree of experimental standardization across the analyzed tumors. Although different staining conditions may alter the absolute numbers of positive cancers, the relative ranking order of positive tumor types will remain unchanged. The cancer entities with highest prevalence and also highest levels of MSLN expression included all types of ovarian and endometrium carcinomas, pancreatic adenocarcinoma, malignant mesothelioma, and adenocarcinomas of the lung, stomach, esophagus, and the colorectum. High rates of MSLN expression have already been described for these tumor entities, although the results varied between studies. Previously described high MSLN positivity rates range from 55% to 100% in ovarian cancer [35,43,44,53], 59% to 76% in endometrium cancer [5,8], 57% to 100% in pancreatic adenocarcinoma [43,47,54], 45% to 100% in malignant mesothelioma [33,43,55,56], 38% to 69% of lung adenocarcinomas [37,38,39], 45% to 78% in stomach cancer [5,45,57,58], 29% to 46% in esophageal adenocarcinoma [6,59], and 30% to 68% of colorectal adenocarcinoma [5,8]. These cancers with a positivity rate of 40% or higher in our study appear to be the best candidates for targeted anti-MSLN therapy. Squamous cell carcinomas of various different sites of origin represent the next group of cancers that show MSLN expression at a relatively high frequency (10–40%). Clinical studies using adaptive immunotherapy (CAR-T cells), monoclonal antibodies (Amatuximab/MORAb-009), recombinant immunotoxins (SS1P, LMB-100/RG7787), antibody-drug conjugates (Anetumab, DMOT4039A), or listeria monocytogene induced anti-tumor immune response (CRS-207) therapies have so far focused on malignant mesothelioma, pancreatic adenocarcinoma, and carcinomas of the ovarian, lung, and breast [24,25,26,27,28,35,60,61,62,63]. 

It is of note that MSLN was initially suggested to represent a diagnostic marker for identification of tumors derived from the mesothelium [1]. Subsequent studies have however identified numerous other tumor entities with MSLN expression [5,6,7,8,9,22,33,35,36,37,38,39,40,41,42,43,44,45,46,47,53,54,55,56,57,58,59,64,65,66,67,68,69,70,71,72,73,74,75,76,77,78,79,80,81,82,83,84,85,86,87,88,89,90,91,92,93,94,95,96,97,98,99,100,101,102,103,104]. That 66 of our 122 analyzed tumor entities contained MSLN positive cases demonstrates that a positive MSLN expression cannot be viewed as an argument for a specific tumor entity. However, a positive MSLN immunostaining might be considered an argument against a tumor origin from organs that never or very rarely gave rise to MSLN positive cancer cells, such as the prostate, thyroid, kidney, germ cell tumors, adrenal tumors, melanoma, many soft tissue tumor types, and hematologic neoplasms.

Our analysis of 1619 colorectal carcinomas identified significant associations with advanced pT stage and lymph node metastases. This is in line with data from 4 earlier studies all describing associations between high MSLN expression and parameters for cancer aggressiveness or poor patient prognosis [42,99,100,104]. However, the absence of statistical associations between MSLN expression and tumor phenotype and/or prognosis in carcinomas of the bladder, breast, ovary, endometrium, kidney, lung, and stomach argues against a major prognostic role of MSLN expression levels. In principle, this notion is consistent with the existing literature. Only 9 previous studies have earlier described associations between high MSLN expression and poor prognosis and/or unfavorable tumor phenotype in these tumor types [43,45,46,57,74,96,105,106,107], while there were 7 other studies which could not find associations with clinico-pathological parameters [5,44,47,58,64,65,92]. The concept of MSLN expression not representing a universal parameter of malignancy is also supported by the frequent MSLN expression in various benign tumors, including Brenner tumors of the ovary, as well as Warthin tumors, pleomorphic adenomas, and basal cell adenomas of the salivary glands. It is of note that high MSLN expression was linked to several key molecular features in the cancer types analyzed, such as triple negative breast cancer and RAS (KRAS or NRAS) mutations. This observation fits with known interactions of MSLN with relevant molecular pathways interacting with these molecular features, such as the MAPK/ERK pathway (interacting with MMP-7, MUC16, and ERK) [15,16,108], deregulation of HER2 expression [109], and the PI3K/AkT pathway (interacting with PI3K and MUC6) [16,108,110]. Importantly, all prevalences described in this study are specific to the reagents and the protocol used in our laboratory. It is almost certain that the use of different antibodies, protocols, and interpretation criteria have jointly caused highly diverse literature data on MSLN expression in cancer (summarized in Figure 5). It is well known that different antibodies designed for the same target protein can vary to a large extent in their binding properties and that protocol modifications greatly impact the rate of immunostained cases [111].

## 5. Conclusions

Our analysis of 12,679 cancers generated a ranking order of cancers according to their frequency of MSLN expression. Top ranked tumor entities, such as ovarian carcinomas, endometrium carcinomas, pancreatic adenocarcinomas, and malignant mesothelioma, thus, may be the best candidates for therapy with drugs targeting MSLN. Despite a link between MSLN positivity and aggressive colon cancer phenotype, the prognostic impact of MSLN expression appears to be low in many other tumor types. This section is not mandatory but can be added to the manuscript if the discussion is unusually long or complex.

## Figures and Tables

**Figure 1 biomedicines-09-00397-f001:**
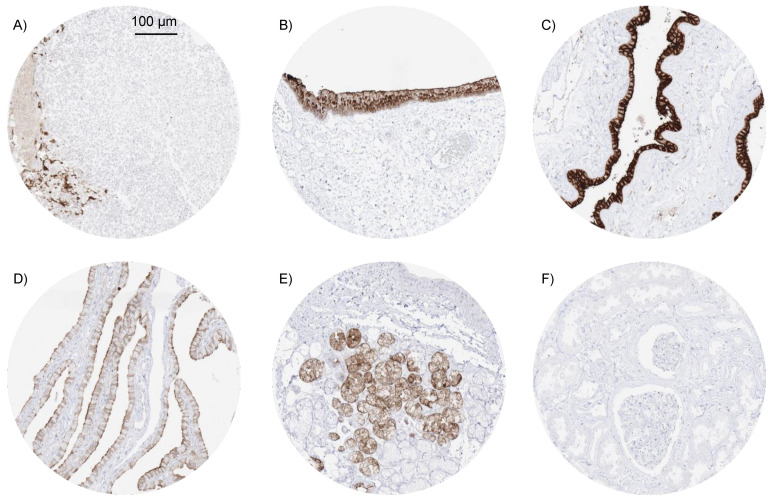
Mesothelin (MSLN) immunostaining in normal tissues. Among normal tissues, MSLN immunostaining is particularly strong in a fraction of squamous epithelial cells in tonsil crypts (**A**), the anal transitional epithelium (**B**) and amnion cells (**C**). A somewhat weaker MSLN staining is seen at the apical cell border and in cilia of fallopian tube epithelium (**D**) and in a fraction mucinous cells (often grouped together) in sublingual glands (**E**). MSLN immunostaining was consistently lacking in the kidney (**F**).

**Figure 2 biomedicines-09-00397-f002:**
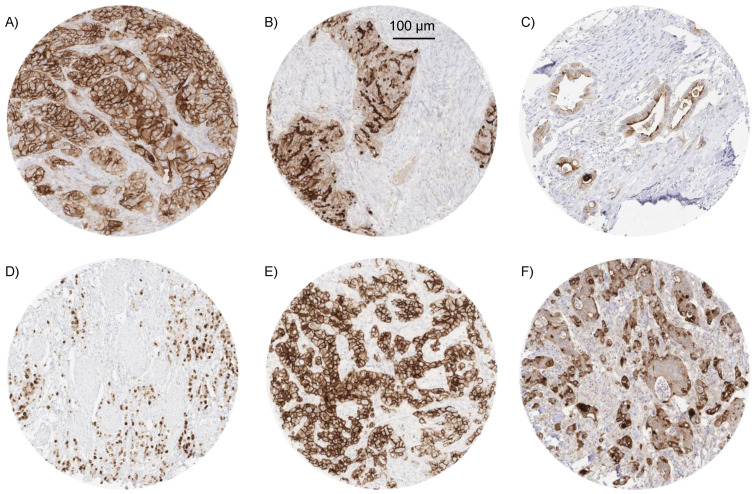
Pattern of MSLN immunostaining in cancer. MSLN immunostaining was found to be strong and predominantly membranous (**A**) or predominantly situated at the apical/luminal cell border (**B**) in two serous high-grade ovarian cancers, variable but predominantly apical in a cholangiocellular carcinoma (**C**), strong and predominantly cytoplasmic in a diffuse type adenocarcinoma of the stomach (**D**), strong and predominantly membranous in a malignant (epitheloid) mesothelioma (**E**), and weak to moderate, cytoplasmic and membranous in a colorectal adenocarcinoma (**F**).

**Figure 3 biomedicines-09-00397-f003:**
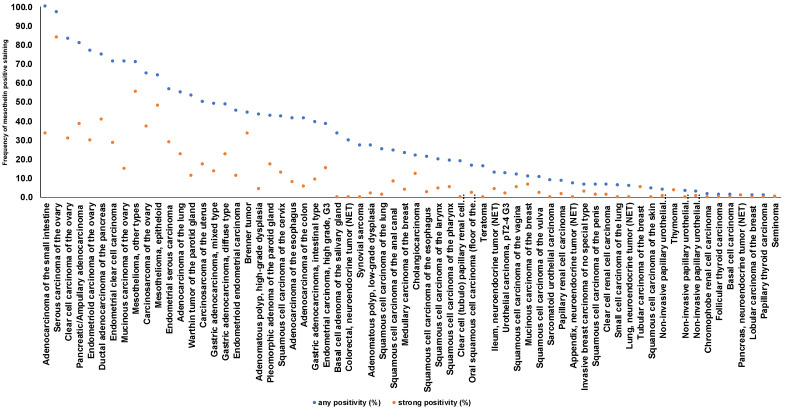
Ranking order of MSLN immunostaining in cancers. Both the frequency of positive cases (blue dots) and the frequency of strongly positive cases (orange dots) are shown. Fifty-six additional tumor entities without any MSLN positive cases are not shown due to space restrictions.

**Figure 4 biomedicines-09-00397-f004:**
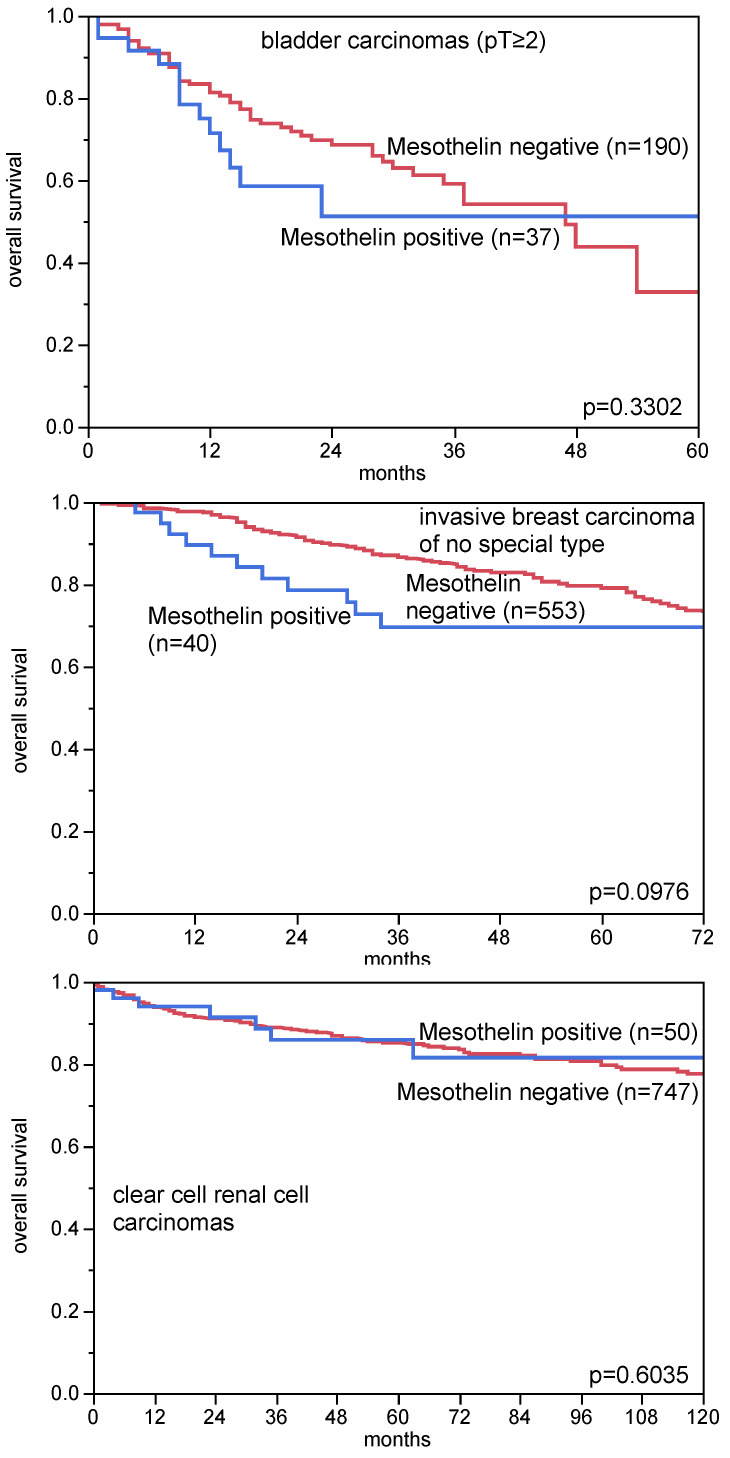
MSLN immunostaining and patient prognosis. All bladder cancer patients had at least pT2 cancers and were treated by cystectomy.

**Figure 5 biomedicines-09-00397-f005:**
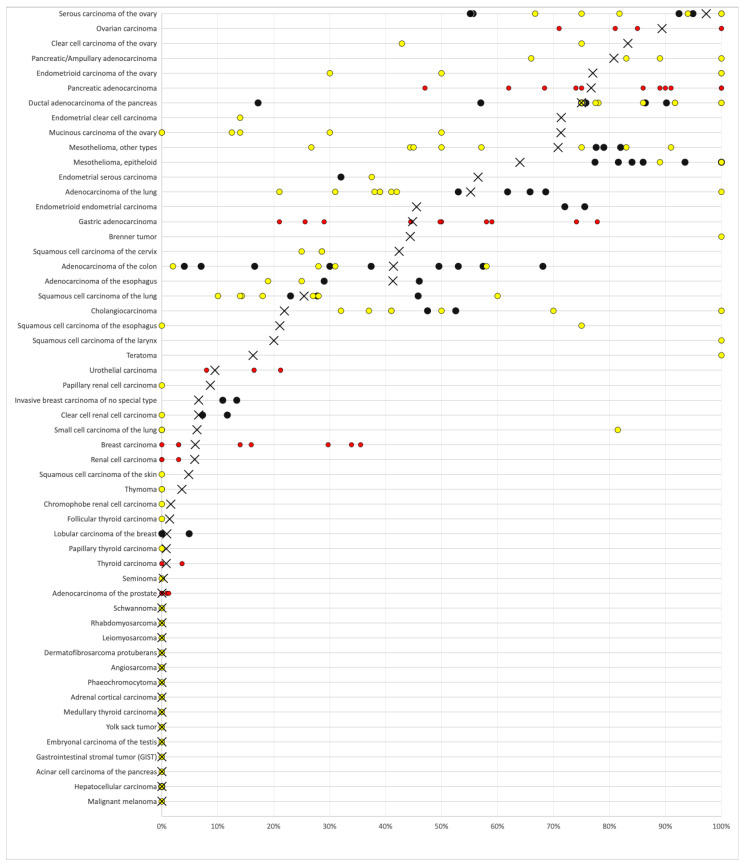
Graphical comparison of MSLN data from this study (×) in comparison with the previous literature. Yellow dots are used for studies involving 1–50 cases, and black dots are used for studies involving more than 50 cases. Red dots are used for studies without subtype analyses. All studies are quoted in the list of references.

**Table 1 biomedicines-09-00397-t001:** MSLN immunostaining in human tumors.

		MSLN Immunostaining						
	Tumor Entity	on TMA (*n*)	Analyzable (*n*)	Negative (%)	Weak (%)	Moderate (%)	Strong (%)	Positive (%)
**Tumors of the skin**	Pilomatrixoma	35	31	100.0	0.0	0.0	0.0	0.0
	Basal cell carcinoma	88	83	98.8	1.2	0.0	0.0	1.2
	Benign nevus	29	26	100.0	0.0	0.0	0.0	0.0
	Squamous cell carcinoma of the skin	90	84	95.2	4.8	0.0	0.0	4.8
	Malignant melanoma	48	44	100.0	0.0	0.0	0.0	0.0
	Merkel cell carcinoma	46	44	100.0	0.0	0.0	0.0	0.0
**Tumors of the head and neck**	Squamous cell carcinoma of the larynx	110	105	80.0	11.4	3.8	4.8	20.0
	Squamous cell carcinoma of the pharynx	60	57	80.7	5.3	8.8	5.3	19.3
	Oral squamous cell carcinoma (floor of the mouth)	130	121	83.5	9.1	5.0	2.5	16.5
	Pleomorphic adenoma of the parotid gland	50	35	57.1	20.0	5.7	17.1	42.9
	Warthin tumor of the parotid gland	49	45	46.7	31.1	11.1	11.1	53.3
	Basal cell adenoma of the salivary gland	15	15	66.7	33.3	0.0	0.0	33.3
**Tumors of the lung, pleura, and thymus**	Squamous cell carcinoma of the lung	127	71	74.6	18.3	5.6	1.4	25.4
	Adenocarcinoma of the lung	250	174	44.8	19.0	13.8	22.4	55.2
	Small cell carcinoma of the lung	20	16	93.8	0.0	6.3	0.0	6.3
	Mesothelioma, epitheloid	39	25	36.0	12.0	4.0	48.0	64.0
	Mesothelioma, other types	76	65	29.2	10.8	4.6	55.4	70.8
	Thymoma	29	28	96.4	0.0	0.0	3.6	3.6
**Tumors of the female genital tract**	Squamous cell carcinoma of the vagina	78	75	88.0	4.0	2.7	5.3	12.0
	Squamous cell carcinoma of the vulva	130	123	89.4	7.3	0.8	2.4	10.6
	Squamous cell carcinoma of the cervix	130	125	57.6	21.6	8.0	12.8	42.4
	Endometrioid endometrial carcinoma	236	220	54.5	22.3	11.8	11.4	45.5
	Endometrial serous carcinoma	82	69	43.5	17.4	10.1	29.0	56.5
	Carcinosarcoma of the uterus	48	46	50.0	21.7	10.9	17.4	50.0
	Endometrial carcinoma, high grade, G3	13	13	61.5	23.1	0.0	15.4	38.5
	Endometrial clear cell carcinoma	8	7	28.6	28.6	14.3	28.6	71.4
	Endometrioid carcinoma of the ovary	115	100	23.0	28.0	19.0	30.0	77.0
	Serous carcinoma of the ovary	567	511	2.7	5.1	8.4	83.8	97.3
	Mucinous carcinoma of the ovary	97	80	28.8	37.5	18.8	15.0	71.3
	Clear cell carcinoma of the ovary	54	42	16.7	33.3	19.0	31.0	83.3
	Carcinosarcoma of the ovary	47	43	34.9	16.3	11.6	37.2	65.1
	Brenner tumor	9	9	55.6	11.1	0.0	33.3	44.4
**Tumors of the breast**	Invasive breast carcinoma of no special type	1391	1138	93.4	2.8	0.8	3.0	6.6
	Lobular carcinoma of the breast	294	215	99.1	0.0	0.9	0.0	0.9
	Medullary carcinoma of the breast	26	26	76.9	11.5	7.7	3.8	23.1
	Tubular carcinoma of the breast	27	19	94.7	0.0	0.0	5.3	5.3
	Mucinous carcinoma of the breast	58	46	89.1	4.3	0.0	6.5	10.9
	Phyllodes tumor of the breast	50	36	100.0	0.0	0.0	0.0	0.0
**Tumors of the digestive system**	Adenomatous polyp, low-grade dysplasia	50	48	72.9	16.7	8.3	2.1	27.1
	Adenomatous polyp, high-grade dysplasia	50	46	56.5	30.4	8.7	4.3	43.5
	Adenocarcinoma of the colon	1882	1186	58.6	29.0	6.9	5.5	41.4
	Adenocarcinoma of the small intestine	10	6	0.0	50.0	16.7	33.3	100.0
	Gastric adenocarcinoma, diffuse type	176	164	51.2	14.6	11.6	22.6	48.8
	Gastric adenocarcinoma, intestinal type	174	170	60.6	18.8	11.2	9.4	39.4
	Gastric adenocarcinoma, mixed type	62	59	50.8	25.4	10.2	13.6	49.2
	Adenocarcinoma of the esophagus	133	75	58.7	14.7	18.7	8.0	41.3
	Squamous cell carcinoma of the esophagus	124	71	78.9	14.1	4.2	2.8	21.1
	Squamous cell carcinoma of the anal canal	91	86	75.6	14.0	2.3	8.1	24.4
	Cholangiocarcinoma	114	105	78.1	4.8	4.8	12.4	21.9
	Hepatocellular carcinoma	50	50	100.0	0.0	0.0	0.0	0.0
	Ductal adenocarcinoma of the pancreas	130	64	25.0	12.5	21.9	40.6	75.0
	Pancreatic/Ampullary adenocarcinoma	58	26	19.2	19.2	23.1	38.5	80.8
	Acinar cell carcinoma of the pancreas	7	6	100.0	0.0	0.0	0.0	0.0
	Gastrointestinal stromal tumor (GIST)	50	43	100.0	0.0	0.0	0.0	0.0
**Tumors of the urinary system**	Non-invasive papillary urothelial carcinoma, pTa G2 low grade	177	154	96.1	3.2	0.0	0.6	3.9
	Non-invasive papillary urothelial carcinoma, pTa G2 high grade	141	125	96.8	3.2	0.0	0.0	3.2
	Non-invasive papillary urothelial carcinoma, pTa G3	187	130	96.9	2.3	0.0	0.8	3.1
	Urothelial carcinoma, pT2-4 G3	940	838	87.5	8.0	2.5	2.0	12.5
	Small cell neuroendocrine carcinoma of the bladder	18	18	100.0	0.0	0.0	0.0	0.0
	Sarcomatoid urothelial carcinoma	25	22	90.9	9.1	0.0	0.0	9.1
	Clear cell renal cell carcinoma	858	807	93.4	3.7	1.6	1.2	6.6
	Papillary renal cell carcinoma	255	242	91.3	5.4	1.7	1.7	8.7
	Clear cell (tubulo) papillary renal cell carcinoma	21	21	81.0	19.0	0.0	0.0	19.0
	Chromophobe renal cell carcinoma	131	124	98.4	0.8	0.8	0.0	1.6
	Oncocytoma	177	170	100.0	0.0	0.0	0.0	0.0
**Tumors of the male genital organs**	Adenocarcinoma of the prostate, Gleason 3 + 3	83	80	100.0	0.0	0.0	0.0	0.0
	Adenocarcinoma of the prostate, Gleason 4 + 4	80	76	100.0	0.0	0.0	0.0	0.0
	Adenocarcinoma of the prostate, Gleason 5 + 5	85	84	100.0	0.0	0.0	0.0	0.0
	Adenocarcinoma of the prostate (recurrence)	330	241	100.0	0.0	0.0	0.0	0.0
	Small cell neuroendocrine carcinoma of the prostate	17	17	100.0	0.0	0.0	0.0	0.0
	Seminoma	624	613	99.7	0.0	0.0	0.3	0.3
	Embryonal carcinoma of the testis	50	44	100.0	0.0	0.0	0.0	0.0
	Yolk sack tumor	50	37	100.0	0.0	0.0	0.0	0.0
	Teratoma	50	43	83.7	11.6	4.7	0.0	16.3
	Squamous cell carcinoma of the penis	80	76	93.4	5.3	0.0	1.3	6.6
**Tumors of endocrine organs**	Adenoma of the thyroid gland	114	104	100.0	0.0	0.0	0.0	0.0
	Papillary thyroid carcinoma	392	358	99.2	0.8	0.0	0.0	0.8
	Follicular thyroid carcinoma	158	146	98.6	1.4	0.0	0.0	1.4
	Medullary thyroid carcinoma	107	91	100.0	0.0	0.0	0.0	0.0
	Anaplastic thyroid carcinoma	45	43	100.0	0.0	0.0	0.0	0.0
	Adrenal cortical adenoma	50	49	100.0	0.0	0.0	0.0	0.0
	Adrenal cortical carcinoma	26	25	100.0	0.0	0.0	0.0	0.0
	Phaeochromocytoma	50	49	100.0	0.0	0.0	0.0	0.0
	Appendix, neuroendocrine tumor (NET)	22	14	92.9	0.0	7.1	0.0	7.1
	Colorectal, neuroendocrine tumor (NET)	10	10	70.0	30.0	0.0	0.0	30.0
	Ileum, neuroendocrine tumor (NET)	49	47	87.2	8.5	0.0	4.3	12.8
	Lung, neuroendocrine tumor (NET)	19	17	94.1	5.9	0.0	0.0	5.9
	Pancreas, neuroendocrine tumor (NET)	102	95	98.9	0.0	0.0	1.1	1.1
	Colorectal, neuroendocrine carcinoma (NEC)	11	9	100.0	0.0	0.0	0.0	0.0
	Gallbladder, neuroendocrine carcinoma (NEC)	4	4	100.0	0.0	0.0	0.0	0.0
	Pancreas, neuroendocrine carcinoma (NEC)	13	12	100.0	0.0	0.0	0.0	0.0
**Tumors of hematopoietic and lymphoid tissues**	Hodgkin Lymphoma	103	101	100.0	0.0	0.0	0.0	0.0
	Non-Hodgkin Lymphoma	62	14	100.0	0.0	0.0	0.0	0.0
	Small lymphocytic lymphoma, B-cell type (B-SLL/B-CLL)	50	50	100.0	0.0	0.0	0.0	0.0
	Diffuse large B cell lymphoma (DLBCL)	114	114	100.0	0.0	0.0	0.0	0.0
	Follicular lymphoma	88	87	100.0	0.0	0.0	0.0	0.0
	T-cell Non Hodgkin lymphoma	24	24	100.0	0.0	0.0	0.0	0.0
	Mantle cell lymphoma	18	18	100.0	0.0	0.0	0.0	0.0
	Marginal zone lymphoma	16	16	100.0	0.0	0.0	0.0	0.0
	Diffuse large B-cell lymphoma (DLBCL) in the testis	16	16	100.0	0.0	0.0	0.0	0.0
	Burkitt lymphoma	5	2	100.0	0.0	0.0	0.0	0.0
**Tumors of soft tissue and bone**	Tenosynovial giant cell tumor	45	43	100.0	0.0	0.0	0.0	0.0
	Granular cell tumor	53	47	100.0	0.0	0.0	0.0	0.0
	Leiomyoma	50	43	100.0	0.0	0.0	0.0	0.0
	Angiomyolipoma	91	84	100.0	0.0	0.0	0.0	0.0
	Angiosarcoma	73	66	100.0	0.0	0.0	0.0	0.0
	Dermatofibrosarcoma protuberans	21	16	100.0	0.0	0.0	0.0	0.0
	Ganglioneuroma	14	14	100.0	0.0	0.0	0.0	0.0
	Kaposi sarcoma	8	6	100.0	0.0	0.0	0.0	0.0
	Leiomyosarcoma	87	84	100.0	0.0	0.0	0.0	0.0
	Liposarcoma	132	116	100.0	0.0	0.0	0.0	0.0
	Malignant peripheral nerve sheath tumor (MPNST)	13	12	100.0	0.0	0.0	0.0	0.0
	Myofibrosarcoma	26	26	100.0	0.0	0.0	0.0	0.0
	Neurofibroma	117	113	100.0	0.0	0.0	0.0	0.0
	Sarcoma, not otherwise specified (NOS)	75	73	100.0	0.0	0.0	0.0	0.0
	Paraganglioma	41	40	100.0	0.0	0.0	0.0	0.0
	Primitive neuroectodermal tumor (PNET)	23	19	100.0	0.0	0.0	0.0	0.0
	Rhabdomyosarcoma	7	7	100.0	0.0	0.0	0.0	0.0
	Schwannoma	121	111	100.0	0.0	0.0	0.0	0.0
	Synovial sarcoma	12	11	72.7	27.3	0.0	0.0	27.3
	Osteosarcoma	43	36	100.0	0.0	0.0	0.0	0.0
	Chondrosarcoma	39	18	100.0	0.0	0.0	0.0	0.0

**Table 2 biomedicines-09-00397-t002:** MSLN immunostaining and tumor phenotype.

			MSLN Immunostaining				*p* Value
		Analyzable (*n*)	Negative (%)	Weak (%)	Moderate (%)	Strong (%)	
**endometrioid endometrial carcinoma**	all cancers	171	55.0	20.5	12.9	11.7	
	pT1	108	55.6	17.6	14.8	12.0	0.5077
	pT2	24	58.3	25.0	4.2	12.5	
	pT3-4	35	51.4	28.6	14.3	5.7	
	pN0	50	42.0	24.0	22.0	12.0	0.6098
	pN+	30	56.7	20.0	13.3	10.0	
**serous high grade ovarian carcinoma**	all cancers	386	3.1	5.2	8.8	82.9	
	pT1	32	0.0	12.5	9.4	78.1	0.3821
	pT2	42	2.4	7.1	4.8	85.7	
	pT3	259	2.7	3.9	9.3	84.2	
	pN0	81	1.2	3.7	9.9	85.2	0.0621
	pN1	166	1.8	6.6	10.8	80.7	
**Invasive breast carcinoma of no special type**	all cancers	1072	93.7	2.7	0.8	2.8	
	pT1	545	95.0	1.8	1.1	2.0	0.0262
	pT2	406	93.8	2.5	0.5	3.2	
	pT3-4	79	86.1	8.9	0.0	5.1	
	G1	150	98.0	1.3	0.0	0.7	<0.0001
	G2	540	97.6	0.7	0.4	1.3	
	G3	381	86.4	6.0	1.8	5.8	
	pN0	544	92.5	2.5	1.8	3.3	0.0390
	pN+	398	95.4	1.7	0.2	2.8	
	non triple negative	737	98.2	0.9	0.1	0.7	<0.0001
	Triple negative	133	67.7	14.3	6.0	12.0	
**Urinary bladder carcinoma**	all cancers	925	90.7	5.8	1.8	1.6	
	pTa G2 low	154	96.1	3.2	0.0	0.6	<0.0001
	pTa G2 high	125	96.8	3.2	0.0	0.0	
	pTaG3	130	96.9	2.3	0.0	0.8	
	pT ≥2 G3	792	87.8	8.1	2.3	1.9	
	pN0	312	87.2	7.4	2.9	2.6	0.7125
	pN+	170	83.7	9.0	4.5	2.8	
**Clear cell renal cell carcinoma**	all cancers	757	93.4	4.0	1.6	1.1	
	pT1	450	93.3	4.0	1.3	1.3	0.8197
	pT2	82	95.1	2.4	1.2	1.2	
	pT3-4	219	92.7	4.6	2.3	0.5	
	ISUP 1	241	92.9	3.7	2.5	0.8	0.6913
	ISUP 2	250	92.0	4.8	1.2	2.0	
	ISUP 3	211	95.3	3.3	0.9	0.5	
	ISUP 4	45	93.3	4.4	2.2	0.0	
	pN0	127	93.7	3.1	2.4	0.8	0.5127
	pN+	19	100.0	0.0	0.0	0.0	
**Gastric carcinoma**	all cancers	373	55.0	19.0	10.2	15.8	
	pT1-2	63	57.1	17.5	12.7	12.7	0.2490
	pT3	122	60.7	18.9	9.0	11.5	
	pT4	122	45.9	22.1	10.7	21.3	
	pN0	83	65.1	13.3	8.4	13.3	0.0697
	pN+	222	48.7	23.0	11.3	17.1	
**Colorectal adenocarcinoma**	all cancers	1619	58.8	28.9	6.5	5.8	
	pT1	68	66.2	30.9	1.5	1.5	<0.0001
	pT2	323	68.4	22.9	5.9	2.8	
	pT3	894	55.8	31.4	7.4	5.4	
	pT4	322	56.5	27.3	5.3	10.9	
	pN0	839	64.2	25.4	5.7	4.6	<0.0001
	pN+	752	52.5	33.0	7.3	7.2	
	MMR proficient	1114	58.6	29.0	6.9	5.5	0.7017
	MMR deficient	82	62.2	25.6	4.9	7.3	
	RAS wildtype	441	62.4	28.1	5.4	4.1	0.0010
	RAS mutation	345	49.0	34.5	9.0	7.5	
	BRAF wildtype	122	60.7	27.9	8.2	3.3	0.1063
	BRAF V600E mutation	21	47.6	19.0	19.0	14.3	
**adenocarcinoma of the lung**	all cancers	174	44.8	19.0	13.8	22.4	
	pT1	83	45.8	19.3	14.5	20.5	0.8172
	pT2	52	46.2	21.2	7.7	25.0	
	pT3	28	42.9	17.9	17.9	21.4	
	pT4	9	44.4	11.1	33.3	11.1	
	pN0	95	51.6	14.7	14.7	18.9	0.1194
	pN1	57	36.8	28.1	10.5	24.6	

Abbreviation: pT: pathological tumor stage, pN: pathological lymph node status, G: grade, ISUP: International Society of Urological Pathology, MMR: mismatch repair.

**Table 3 biomedicines-09-00397-t003:** MSLN immunostaining and HPV status.

	MSLN Immunostaining	*n*	HPV Status		
			Negative (%)	Positive (%)	
All cancers	negative	427	56.7	43.3	0.0098
	positive	93	41.9	58.1	
Oral squamous cell carcinoma	negative	60	85.0	15.0	0.3061
	positive	15	73.3	26.7	
Squamous cell carcinoma of the pharynx	negative	43	44.2	55.8	0.0996
	positive	3	18.2	81.8	
Squamous cell carcinoma of the larynx	negative	47	83.0	17.0	0.4281
	positive	12	91.7	8.3	
Squamous cell carcinoma of the cervix	negative	48	10.4	89.6	0.6182
	positive	28	14.3	85.7	
Squamous cell carcinoma of the vagina	negative	26	50.0	50.0	1.0000
	positive	4	50.0	50.0	
Squamous cell carcinoma of the vulva	negative	71	70.4	29.6	0.0698
	positive	8	20.0	80.0	
Squamous cell carcinoma of the penis	negative	68	38.2	61.8	0.3435
	positive	5	60.0	40.0	
Squamous cell carcinoma of the skin	negative	37	97.3	2.7	0.8161
	positive	1	100.0	0.0	
Squamous cell carcinoma of the anal canal	negative	27	11.1	88.9	0.4237
	positive	9	22.2	77.8	

## Data Availability

All data generated or analyzed during this study are included in this published article.

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
