# Peer review of "Mesothelin Expression in Human Tumors: A Tissue Microarray Study on 12,679 Tumors"

_biomedicines, 2021, doi:10.3390/biomedicines9040397_

Round 1
Reviewer 1 Report
The authors examined the prevalence and significance of mesothelin using Tissue Microarrays.
Overall, the study was well-performed; however, there are several questions and comments as follows.
1) More information on targeted cancer therapies is required in the introduction and discussion.
2) Please pay attention to the terminology of the human or murine gene name
3) The authors did not describe the interobserver agreement for the immunohistochemical stain—a minimum of two pathologists is required to evaluate the immunohistochemical stain.
4) The authors should assess and describe the intratumoral heterogeneity of the immunohistochemical expression.
5) The authors should describe the stat of received neoadjuvant therapy or not.
6) Is there a difference in the expression of mesothelin depending on the center or the invasive front of the tumor in the TMA sample?
7) Scale bars or magnification should be included in micrographs.
Reviewer 2 Report
The authors have done a great job considering this large number of tumor samples to analyze the mesothelin expression
But there are some considerations to keep in mind
Figure 3 makes it difficult to see the samples reflected on the "X" axis. Maybe it could be made bigger or use a code for each sample….
Table 2, the authors forgot to put parentheses to the (n)
In the description of the staining process, the authors should indicate the methodology used: incubation time, type of staining, etc.
Round 2
Reviewer 1 Report
All previous concerns and suggestions have been addressed sufficiently by the authors.